# Towards More Equitable Ulcer Recognition Models: A Dataset of Naturalistic Foot Images from People of Color Living with Diabetes

Cynthia M. Baseman
*School of Interactive Computing*
*Georgia Institute of Technology*
Atlanta, USA
cbaseman3@gatech.edu

Yingtian Shi
*School of Interactive Computing*
*Georgia Institute of Technology*
Atlanta, USA
yshi457@gatech.edu

Zikang Leng
*School of Interactive Computing*
*Georgia Institute of Technology*
Atlanta, USA
zleng7@gatech.edu

Yaqi Liu
*School of Interactive Computing*
*Georgia Institute of Technology*
Atlanta, USA
yliu3387@gatech.edu

Gabriel Santamarina
*School of Medicine*
*Emory University*
Atlanta, USA
gabriel.santamarina@emory.edu

Marcos C. Schechter
*School of Medicine*
*Emory University*
Atlanta, USA
marcos.coutinho.schechter@emory.edu

Maya Fayfman
*School of Medicine*
*Emory University*
Atlanta, USA
maya.fayfman@emory.edu

Thomas Ploetz
*School of Interactive Computing*
*Georgia Institute of Technology*
Atlanta, USA
thomas.ploetz@gatech.edu

Rosa I. Arriaga
*School of Interactive Computing*
*Georgia Institute of Technology*
Atlanta, USA
arriaga@cc.gatech.edu

*Abstract*—Diabetic foot ulcers, a life-threatening complication of diabetes, take a disproportionate toll on communities of color; however, these communities are currently underrepresented in dermatologic and wound image datasets. Further, many of these datasets were collected under controlled conditions, limiting the transferability of ulcer recognition models to naturalistic settings. In support of more equitable and generalizable computational modeling, we detail our two-year effort to create the first repository of diabetic foot ulcer images collected predominantly from patients of color in naturalistic settings. We conduct an evaluation of state-of-the-art foot ulcer segmentation and classification methods using our dataset of 3,362 foot images collected from 252 patients, and provide evidence that current ulcer recognition models result in insufficient performance: the best performing baseline model (Mask R-CNN) has been previously reported to achieve a Dice score of 90.2%, but achieves only 39.5% on our more naturalistic dataset from patients of color. We propose and evaluate a new pipeline which improves segmentation performance, including an ulcer detection model and a foundational segmentation model (Segment Anything 2 UNet) tailored to communities of color and specifically aiming for naturalistic assessment scenarios. We release our image dataset to support the development of larger, more diverse datasets, and ultimately more equitable models for diabetic foot care.

*Index Terms*—Diabetic Foot Ulcer, Computer Vision, Image Classification, Segmentation, Health Equity

This work was supported by the American Diabetes Association Grant 11-22-ICTSHD-09, the GT-Emory AI Humanity seed grant, and NIH award DK124647.

## I. INTRODUCTION

Ten percent of U.S. American adults have a diagnosis of diabetes, and 34% of those with diabetes will experience a diabetic foot ulcer (DFU) within their lifetime [1]. These open foot sores or wounds lead to an infection in 50% of cases, and 20% of DFU patients may require an amputation, with a mortality rate of 70% within 5 years [1]. Foot ulceration is complex, with variable clinical presentation and high recurrence rates. DFU healing is assessed in part through dermatologic indicators: wound depth (to the dermis, fat, muscle, or bone), wound surface area, and signs of infection. Measuring wound area with a ruler is a traditional approach, but it can be tedious and inaccurate [2]. Clinicians are in urgent need of methods to efficiently assess and respond to the complex pathogenesis, slow healing, and high rates of recurrence and complications throughout the treatment course.

With recent advancements, computer vision approaches may offer opportunities to routinely and automatically analyzing images of the foot to segment ulcers and determine clinically relevant characteristics. Researchers are therefore exploring deep learning models for DFU segmentation from images [3]. For example, Wang *et al.* [4] compiled the AZH Wound Care Dataset, consisting of 1,010 close-up images of ulcers. Utilizing convolutional neural networks (CNN) such as Mask R-CNN [5], MobileNetV2 [6], U-Net [7], and SegNet [8], they achieved a high Dice score of 90.5% in the task of DFU

segmentation. Additionally, Liao *et al.* proposed HarDNet-DFUS [9], another CNN-based model that obtained state-of-the-art performance on the DFU challenge (DFUC) 2022 dataset [10] comprised of 4,000 ulcer images.

Although many recent computer vision studies have offered promising results, communities of color[1] are currently under-represented in dermatologic and wound datasets [12], [13]. Diabetes takes a disproportionate toll on communities of color [14], [15] and health disparities may be further propagated by biased databases [16]. The very limited available data for those with darker skin tones presents unique challenges for utilizing current wound recognition models for communities of color. This lack of data prevents a systematic analysis of the generalizability of the state-of-the-art recognition models.

The contribution of this work is three-fold. In support of more generalizable and equitable computational modeling, we publicly release a repository of DFU images collected mostly from patients of color. Our dataset consists of 3,362 foot images collected from 252 participants, including 1,551 instances of a DFU or pre-ulcerative lesion (a callous or ulcer that does not penetrate to the subdermis). We compare our dataset to other available wound datasets regarding dataset composition and data collection techniques. Second, we leverage our dataset to contribute a performance evaluation of state-of-the-art foot ulcer segmentation and classification models on patients of color. Finally, to increase model performance, we propose and evaluate a new pipeline including an ulcer detection model and a foundational segmentation model (Segment Anything 2 UNet) tailored to communities of color and aiming for naturalistic assessment scenarios.

## II. Dataset Construction

We first discuss our over two-year effort to collect the first diabetic foot ulcer (DFU) dataset predominantly from patients of color, including image collection techniques and dataset composition. Our work was approved by two IRBs, including that of the hospital where data collection occurred.

### A. Data Collection Methodology

*1) Image Collection:* For over two years, university researchers collaborated with a clinical team at an urban, safety net hospital in a southern U.S. state. Per year, over 450 patients receive DFU-related care at the hospital-based diabetes clinic, and over 250 patients are hospitalized with a DFU [17]. Most patients (80%) identify as non-Hispanic Black [17].

Foot images were collected by a team of students, researchers, and clinicians. After a participant signed the consent form, foot images were taken in the clinical examination room or other clinical setting. Participants were excluded from image collection if they were under 18 years old, could not read and understand English, or if they had experienced a major amputation. Our dataset does include patients with amputations below the ankle, typically referred to minor

---

[1]In the United States, the term "communities of color" or "people of color" is frequently used within health equity contexts to describe individuals not considered "white" [11].

---

amputations. Participants provided consent at their baseline (time 0) appointments to provide their foot images during clinical appointments for up to 52 weeks. Participants were compensated with $10 cash at their baseline appointment, as well as an additional $10 per month for up to 4 months.

Images were collected using smartphones or iPads and were taken from multiple angles, including dorsal, plantar, medial, lateral, toe tips, and heel (See Fig. 1). The entire foot was included in the photo, and the lighting and background varied. Ground truth data was digitally recorded after clinical examination by a physician or podiatrist, including the presence and location of ulcers and pre-ulcerative lesions.

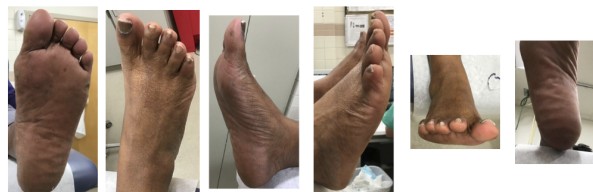

Fig. 1. Six image angles: plantar, dorsal, medial, lateral, toe tips, and heel.

*2) Dataset Annotation:* A team of researchers annotated the images using AnyLabeling software [18]. Boundaries were first annotated for the foot. Segment Anything (SAM) [19] was used to assist foot boundary annotation, but segmentation was manually adjusted as needed to ensure accuracy. Referencing the ground truth collected in-clinic, boundaries were then annotated for ulcers and pre-ulcerative lesions.

To increase annotation reliability, each image was reviewed by a second annotator and annotation discrepancies were discussed. Throughout the process, the research team held recurring meetings with a podiatrist to verify annotation accuracy, during which images from most (over 50%) of the patients with ulcers and/or pre-ulcerative lesions were discussed. To respect the podiatrist's limited availability and prioritization of clinical tasks (especially due to their affiliation with a safety net hospital), patients were only discussed when image annotators had questions. Two co-authors each completed a final check of the entire dataset to ensure agreement.

In addition, the research team manually reviewed images for any anonymity concerns to protect participant privacy. Identifiable information including faces, names, and distinctive tattoos were redacted from the dataset. Redacted areas were annotated as such using AnyLabeling, to assist future researchers in excluding these areas from modeling efforts.

*3) Dataset Composition:* In total, 3,362 foot images were collected from 252 patients (252 baseline appointments and 126 follow-ups). The number of images from each patient varied, ranging from 3 (only baseline, with a major amputation) to 49 images (multiple follow-ups). The average patient age at baseline was 57 years and the median was 58 years. 62% of the patients were male and 38% were female.

As shown in Table I, the image dataset includes the following annotation instances: 882 DFUs (including surgical wounds) and 669 pre-ulcerative lesions. Multiple annotations

TABLE I
OVERVIEW OF OUR DATASET (N=252 PARTICIPANTS)

| | Total Instances | Images (N=3362) | |
|---|---|---|---|
| Ulcer or Surgical Wound | 882 | 751 | 22.3% |
| Pre-Ulcerative Lesion | 669 | 467 | 13.9% |
| Area of Concern (Ulcer or Lesion) | 1551 | 1135 | 33.8% |

may be present in one image, and a wound may appear in more than one photographed angle. Ulcers and pre-ulcerative lesions were prevalent on the plantar (34% and 39%, respectively), medial (21% and 20%), dorsal (15% and 14%), lateral (16% and 18%), toe tips (11% and 8%), and heel (3% and 1%).

The image repository can be requested at https://github.com/tploetz/dfu-recognition.

### B. Comparison to Previous Datasets

Wound image collection and research has been supported by challenges such as the DFU challenge [10], [20]. There are now several DFU datasets available, ranging from 100 images or less [21], [22] to thousands of images [10], [23].

Our dataset is the first known DFU dataset predominantly collected from patients of color. Most previous studies do not report the race or ethnicity of their participants and the associated publications feature only images of white or pale skin [4], [9], [10], [21], [24]–[26]. As communities of color experience more severe diabetes complications [14], [15], their equitable representation within wound datasets is imperative.

Further, an aim of automated wound segmentation is to allow for remote monitoring and increased patient engagement. Images taken under controlled settings may not translate into models that perform accurately on images taken in naturalistic settings, such as in a patient's own home. Some previous wound datasets have been collected using an image capture box with controlled lighting, angle, and distance to the foot [21], [24]. Such methods which only capture the plantar view may fail to identify foot complications on other regions of the foot [27]. In our dataset, for example, 15% of the ulcers photographed were on the dorsal region. Additionally, image datasets which were taken by digital cameras [4], [10], [23] may not be accessible to safety net hospitals. The images in our dataset were taken exclusively with smartphones or iPads.

Two final distinctions of our dataset are that we capture the entire foot and include control images, i.e., images in which no ulcer or pre-ulcerative lesion is present. Many previous studies only included close-up images of wounds [4], [10], [22]–[24]. A system intended for patient self-monitoring and clinical decision support must process the whole foot and identify areas of concern, as opposed to requiring a patient or caregiver to pre-assess the foot to identify such areas.

Compared to previous datasets, we believe our repository is more representative of how real clinicians or patients would capture foot images. This leads to a dataset with potentially confounding factors influencing performance, but provides a more realistic benchmark for eventual clinical impact.

## III. METHODOLOGY

We now describe our approach for automatic ulcer recognition from images, leveraging an ulcer detection model and a segmentation model tailored for communities of color. We then describe our experimental set-up.

### A. Proposed Ulcer Recognition Pipeline

Our proposed pipeline (see Fig. 2) for automatic DFU recognition integrates a segmentation model (Segment Anything 2 [28] UNet), with an ulcer detection model (YOLOv12-X [29]) which predicts bounding boxes for ulcers in images. We treat the region within each bounding box as an area of interest potentially containing an ulcer, and therefore assign higher weights to pixels within the predicted bounding boxes in the SAM2-UNet output. After weighting, we apply a dynamic thresholding technique based on the maximum and minimum pixel confidence scores in the image to generate the final segmentation output. Below we provide further details on the ulcer detection model and segmentation model.

### Ulcer Detection Model: YOLOv12-X

We employ YOLOv12-X [29] as our ulcer detection model. With 59.3 million parameters, YOLOv12-X is a relatively lightweight CNN-based model that offers state-of-the-art object detection performance. YOLOv12-X introduced the attention mechanism based on the original YOLO model, which helps the model better capture local information of the image, achieving more effective object detection. Its size and efficiency make it particularly well-suited for our dataset, as it minimizes the risk of overfitting that can occur with more complex vision transformer-based models, especially when trained on smaller datasets. We employed the SGD (Stochastic Gradient Descent) optimizer with a learning rate of $10^{-2}$ and a weight decay of $10^{-2}$ for optimization. The model was trained for 30 epochs with a batch size of 16. We followed the default fine-tune loss function from YOLOv12 as shown in Equation 1, where $\mathcal{L}_{\text{box}}$ is the bounding box regression loss, $\mathcal{L}_{\text{cls}}$ is the classification loss, and $\mathcal{L}_{\text{dfl}}$ is the distribution focal loss. N is the number of predictions, and $\mathbb{1}_i^{\text{obj}}$ is an indicator function that equals 1 if an object exists in box i, and 0 otherwise. The term $\text{CIoU}(\hat{b}_i, b_i)$ represents the Complete IoU between the predicted bounding box $\hat{b}_i$ and the ground-truth box $b_i$. The variables $c_i$ and $\hat{c}_i$ denote the ground-truth and predicted class probabilities, respectively. For the Distribution Focal Loss, $t_l^j$ and $t_r^j$ are the left and right integer targets for each coordinate j, and $\hat{d}_i^j$ is the predicted distribution for coordinate j. The weights $w_l^j$ and $w_r^j$ are calculated as $w_l^j = t_r^j - t^j$ and $w_r^j = 1 - w_l^j$. K represents the number of coordinates (four for x, y, w, h). The term CE represents Cross Entropy loss. The terms $\lambda_{\text{box}}, \lambda_{\text{cls}}, \lambda_{\text{dfl}}$ denote the corresponding loss weights for

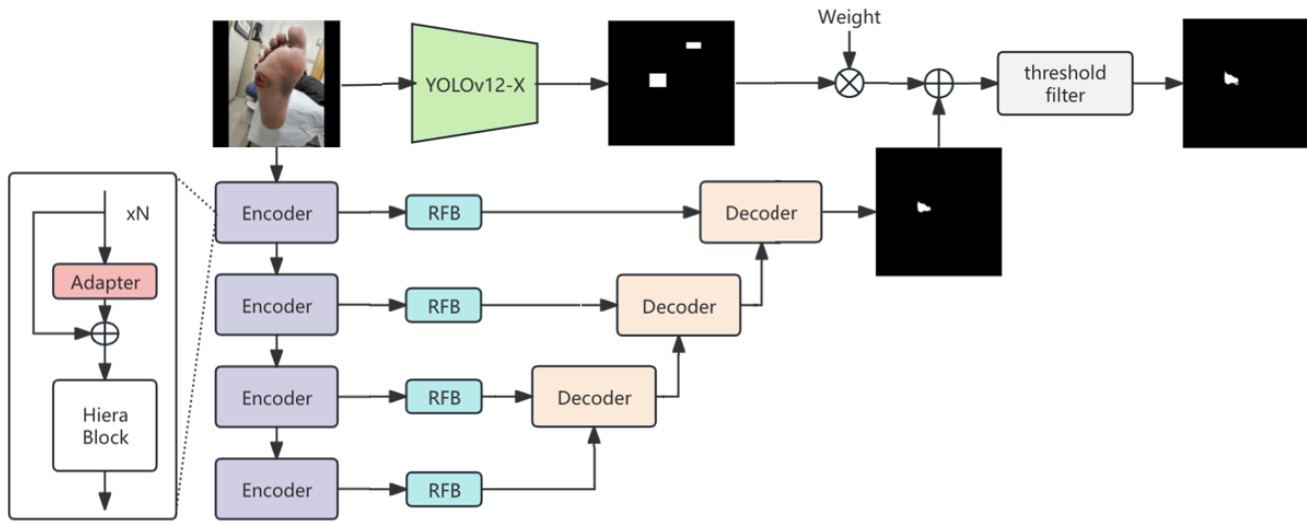

Fig. 2. Overview of the proposed system. For classification, YOLOv12-X detects ulcers and outputs regions of interest. For segmentation, these are then combined with a fine-tuned SAM2-UNet for mask prediction. For additional details on YOLOv12-X and SAM2-UNet, please refer to [29] and [28], respectively.

each component, set to 7.5, 0.5, and 1.5, respectively.

$$
\begin{aligned}
\mathcal{L} = &\ \lambda_{\text{box}} \underbrace{\sum_{i=1}^{N} \mathbb{1}_i^{\text{obj}} \cdot \left( 1 - \text{CIoU}(\hat{b}_i, b_i) \right)}_{\mathcal{L}_{\text{box}}:\text{CIoU loss}} \\
&+ \lambda_{\text{cls}} \underbrace{\sum_{i=1}^{N} \left[ -c_i \log(\hat{c}_i) - (1 - c_i) \log(1 - \hat{c}_i) \right]}_{\mathcal{L}_{\text{cls}}:\text{BCE classification loss}} \\
&+ \lambda_{\text{dfl}} \underbrace{\sum_{i=1}^{N} \mathbb{1}_i^{\text{obj}} \cdot \frac{1}{K} \sum_{j=1}^{K} (w_l^j \cdot \text{CE}(\hat{d}_i^j, t_l^j) + w_r^j \cdot \text{CE}(\hat{d}_i^j, t_r^j))}_{\mathcal{L}_{\text{dfl}}:\text{Distribution Focal Loss}}
\end{aligned}
$$

(1)

**Segmentation Model: SAM2-UNet**

Medical image instance segmentation has traditionally been tackled using end-to-end trained models, where all components are trained simultaneously. Deep learning-based models have been highly effective due to their proficiency in learning intricate image features [7], [30], [31]. These models, however, typically specialize in specific segmentation tasks and require retraining for different applications, resulting in a "specialist model." Such task-specific design is a significant limitation, as model performance can degrade when applied to new tasks or types of data. This lack of versatility stands in contrast to recent advances in natural image segmentation, such as the foundational Segment Anything Model (SAM) [19]. Such models demonstrate remarkable flexibility and effectiveness across various segmentation challenges, offering a stark contrast to the more limited, specialist models commonly used in medical imaging. While SAM has been applied to other domains such as brain MRI and cancer segmentation [32], [33], to our knowledge it has not yet been applied to DFU.

As a successor to SAM, Segment Anything 2 (SAM2) [28] utilizes a bigger dataset and has improved architectural design. For example, MedSAM-2 achieved efficient organ segmentation on medical CT images [34]. In the absence of manual prompts, however, SAM2 still yields class-agnostic segmentation results. To address this limitation, SAM2-UNet builds upon SAM2's hierarchical backbone by introducing a more efficient U-shaped architecture. This design not only preserves the model's ability to generalize across diverse image types but also enhances segmentation performance through the well-established advantages of U-shaped networks [35]. SAM2-UNet has demonstrated its effectiveness on tasks such as polyp segmentation [35].

SAM2-UNet consists of four components: image encoder, decoder, receptive field blocks (RFBs), and adapters. The image encoder utilizes the Hiera backbone pretrained by SAM2 to transform the input image into high-dimensional embeddings. Following feature extraction, four receptive field blocks are employed to reduce the channel dimensions to 64 while simultaneously enhancing lightweight features. Due to the large number of parameters in Hiera, SAM2-UNet freezes its weights and inserts adapters before each multi-scale block to enable parameter-efficient fine-tuning. Finally, the mask decoder adopts a customizable U-shaped architecture. The output from each decoder block is passed through a segmentation head to generate the final segmentation result.

Fine-tuning SAM on a downstream dataset can enhance its performance in medical segmentation tasks [32]. We fine-tuned SAM2-UNet on our dataset, using the sum of binary cross entropy loss and dice loss as our loss function, as shown in Equation 2. In Equation 2, $y_i$ and $p_i$ represent the ground truth and predicted segmentation for pixel $i$, respectively, and $N$ is the total number of pixels. We employed the AdamW optimizer with a learning rate of $10^{-4}$ and a weight decay of

$10^{-2}$ for optimization. The model was trained for 20 epochs with a batch size of 12, following previous work [32].

$$L = -\frac{1}{N} \underbrace{\sum_{i=1}^{N} [y_i \log(p_i) + (1 - y_i) \log(1 - p_i)]}_{L_{\text{BCE}}}$$

$$+ \underbrace{1 - \frac{2 \sum_{i=1}^{N} p_i y_i}{\sum_{i=1}^{N} p_i^2 + \sum_{i=1}^{N} y_i^2}}_{L_{\text{Dice}}}. \quad (2)$$

### B. Experimental Setup

Our first set of experiments contribute an evaluation of both state-of-the-art foot ulcer recognition models and our proposed pipeline on our dataset of wounds predominantly collected from patients of color. We then evaluate our proposed pipeline's performance on currently available datasets, which consist mostly of images of white or pale feet.

For DFU prediction and prevention, recognizing pre-ulcerative lesions is just as important as recognizing ulcers. We therefore consider both ulcers and lesions as "areas of concern" and aim to detect and segment these areas from images. Most participants presented with areas of concern; however, these areas are not visible in all images due to varying capture angles. Consequently, only 34% of our images contain an area of concern, leading to an imbalanced dataset.

To balance the dataset and enhance training data, we apply transformation-based data augmentation. Transformations included rotation (90, 180, 270 and ±25 degrees random rotation, applied with 0.8 probability), flipping (horizontal flipping and vertical flipping with 0.8), zoom (scaling between 0.8x–1.2x with 0.8), distortion (0.5 with mild warping), and shear (applied with 0.8 probability) while keeping the testing dataset unaugmented for fair evaluation. All images are resized to a resolution of 640 × 640 pixels for input consistency.

*1) Evaluation of Models & Proposed Pipeline on Our Dataset:* In order to establish a baseline, we train and evaluate five state-of-the-art models that have been previously evaluated on the DFUC 2022 dataset [10] and the AZH Wound Care dataset [4]. The models are HarDNet-DFUS [9], Mask R-CNN [5], MobileNetV2 [6], U-Net [7], and SegNet [8].

We randomly split our dataset by participant and run a five-fold evaluation. We used grid search to determine the learning rate and weight decay. The learning rate varied from $10^{-6}$ to $10^{-3}$ and the weight decay varied from $10^{-4}$ to $10^{-2}$.

In addition to training the models on our dataset, we evaluated versions of the HarDNet-DFUS model pretrained on the DFUC2022 dataset. We obtained the open-source model weights and applied them to evaluate that model's performance on our dataset, both directly (without further training) and with fine-tuning on our dataset. This approach allowed us to compare the effectiveness of the model when trained on our dataset versus its performance with the knowledge it had already acquired from the DFUC2022 dataset.

TABLE II
MODEL PERFORMANCE WHEN TRAINED AND EVALUATED ON DFUC2022 AND AZH WOUND CARE DATASETS. RESULTS FROM [10] AND [4]

| | HarDNet-DFUS | Mask R-CNN | MobileNetV2 | U-Net | SegNet |
|---|---|---|---|---|---|
| Dataset | DFUC2022 | AZH Wound Care | | | |
| Dice | 72.9 | 90.2 | 90.3 | 90.2 | 85.1 |
| IoU | 62.5 | (Not Provided in [4]) | | | |

### Classification Task

We first consider the problem of detecting whether there is an area of concern (i.e., ulcer or pre-ulcerative lesion), as not all of our images contain such areas. Simply providing binary information (presence or absence) regarding a foot complication can be informative for patients, serving as a sign to consult a clinician. For this classification problem, we use precision, recall, and the binary F1 score as evaluation metrics.

### Segmentation Task

While the classification task requires only a binary assessment of an image (i.e., area(s) of concern is absent or present), the segmentation task requires not only detecting all areas of concern but also forming accurate boundaries for each of these areas. If a foot has multiple areas of concern, accurate classification requires only recognizing one of them, while accurate segmentation would require recognizing and forming accurate boundaries for all of them. We employ the commonly used Dice Similarity Index and Intersection Over Union (IoU) as evaluation metrics for segmentation performance. However, we note that all of the images in the DFUC2022 dataset and the AZH Wound Care dataset contain ulcers. To establish a fair comparison of models trained on different datasets, we therefore only evaluate the segmentation results on the images within our testing set which include an area of concern.

*2) Evaluation of Proposed Pipeline on Previous Datasets:* We also evaluate our propose pipeline's performance on three currently available datasets, which consist mostly of images of white or pale feet. We trained our model on the FUSeg dataset [4], AZH Wound Care dataset [4], and Medetec dataset [36], and compare the results.

## IV. EXPERIMENTAL RESULTS

### A. Evaluation of Models & Proposed Pipeline on Our Dataset

Tables II and III show the results of the five state-of-the-art models when trained and evaluated on previous datasets and our dataset predominantly collected from patients of color, respectively. For all five models, the Dice scores and IoU achieved on our dataset are significantly lower compared to those achieved on the DFUC2022 and the AZH Wound Care datasets. This observation indicates that our dataset may be more challenging to model due to factors such as skin tone, naturalistic settings, and areas of concern (i.e., ulcers or pre-ulcerative lesions) which are a small region of the image. In the DFUC2022 dataset, 34% of the ulcers are less than 0.5% of the image size, whereas in our dataset (which captures the entire foot), the majority of the areas of concern are less than 0.5% of the image size. These relatively small areas of concern

present a significant challenge, as deep learning segmentation models are known to struggle in detecting small regions [37].

Table III also shows the performance of our proposed pipeline (detailed in Section III-A and Fig. 2). All models were trained and evaluated on our dataset, except for "HarDNet-DFUS (Pretrain)," which was first trained on the DFUC2022 dataset, then was evaluated on our dataset. In Table III, Dice and IoU scores are obtained by evaluating the model solely on images within our testing dataset that contain an area of concern. Our model outperforms all baselines across both classification and segmentation. Specifically, it achieves the highest classification precision (86.8%), recall (83.4%), and F1 score (85.1%), as well as the best segmentation performance with a Dice of 54.4% and IoU of 45.4%. Our proposed pipeline results in relative Dice performance improvements ranging from 37.7% (Mask R-CNN) to 197.3% (SegNet).

We conducted focused ablation analyses on the most critical components of our pipeline: YOLOv12-X and the dynamic thresholding module. YOLO plays an essential role in suppressing false positives by preventing mask generation on ulcer-free images. While its presence slightly reduces the Dice score on ulcerated regions due to its bounding box focus, this trade-off is necessary for robust overall performance. The thresholding module enhances results by effectively filtering noise, particularly low-confidence regions along ulcer boundaries and small isolated artifacts.

An empirical review of model errors provides further insights (see Fig. 3). Classification false positives primarily occurred due to abnormal nails, minor amputations, or calluses. Indirect angles of capture occasionally led to false negatives. Segmentation errors occurred when the model struggled with precise edge delineation, e.g., for ulcers with gradual transitions or irregularly shaped ulcers. The model sometimes over-segmented by including peri-wound skin, and complete misses occurred most often for small (less than 1cm) or atypically-colored ulcers. We believe our work provides a path towards applications for wound size measurement and screening purposes; however, our findings highlight the need for improved edge detection and skin tone adaptation.

### B. Evaluation of Proposed Pipeline on Previous Datasets

We also evaluate our proposed pipeline's performance on three accessible foot ulcer datasets: FUSeg [4], AZH [4], and Medetec [36]. Table IV highlights the cross-dataset generalization performance of our segmentation model, measured using mean Dice coefficient (mDice), when trained and tested on these publicly available ulcer datasets. In Table IV, the "Combination" dataset is the combination of these three available datasets, totaling just under 2,500 images.

An important trend observed is the poor generalization performance on our dataset when models are trained on these other datasets consisting mostly of images of white or pale feet. For example, training on FUSeg or AZH yields mDice scores of just 27.0% and 18.9%, respectively, when tested on our dataset, indicating a significant domain gap. Even training on the "Combination" set, which performs well on

other benchmarks, results in only 20.8% mDice on our dataset. In contrast, a model trained specifically on our dataset achieves better generalization to other datasets, with scores ranging from 70.7% (FUSeg) to 84.0% (Medetec). This asymmetry highlights the unique challenges and visual characteristics present in our dataset, which contains more naturalistic images from patients of color. These findings reinforce the need for inclusive training data and further motivate our design of models tailored to populations underrepresented in dermatologic and wound image datasets.

## V. DISCUSSION AND CONCLUSION

Communities of color are disproportionately impacted by diabetes and DFUs [14], [15] but underrepresented in wound image datasets. This has prevented systematic generalizability analyses of ulcer recognition models. Our experimental findings show that existing computer vision methods do not achieve state-of-the-art results for these communities. To support more equitable computational modeling, we created and released the first known repository of naturalistic DFU images predominantly collected from patients of color. By evaluating state-of-the-art DFU recognition models on this dataset and proposing a new pipeline, our work shows that concerted dataset collection with a particular community can increase model performance for that community.

The evaluation provided in this work reinforces that the impact of skin color is not only skin deep: future work in wound recognition must actively recruit patients of color. Researchers should also collect longitudinal images and expand the scope of their models to segment "areas of concern," including pre-ulcerative lesions. This would also allow for predictive work with the aim of limb preservation, which is especially important for under-resourced settings. An ulcer recognition model could be deployed on a mobile phone, for example to enhance patient engagement and enable remote wound self-monitoring [3], [38]. More equitable detection models may also be deployed in safety net hospitals to assist health care professionals in triaging patients, ensuring that those who need immediate intervention are prioritized [38].

Our work has limitations. The performance of our proposed pipeline still falls short compared to the models trained on previous datasets of white or pale feet. Researchers must probe the causes of inequities and underrepresentation, and advocate for the multi-institutional collection of a much larger dataset. To increase model performance, future researchers might also consider methods such as post-processing refinements, generative adversarial networks, self-supervised learning, and incorporating multi-angle information. XAI techniques (e.g., Grad-CAM) could help visualize decision-critical features across skin tones and conduct rigorous subgroup fairness analysis. In addition, our dataset was predominantly collected from Black patients. Our work provides evidence that state-of-the-art DFU recognition models are not generalizable to all skin tones. We hope our work will serve as a call to action for researchers to work towards more equitable models, including working with other underrepresented communities of color to collect

TABLE III
MODEL PERFORMANCE WHEN TRAINED AND EVALUATED ON OUR DATASET

| | Precision | Recall | F1 | Dice | IoU |
|---|---|---|---|---|---|
| | Classification | | | Segmentation | |
| HarDNet-DFUS (Pretrained on DFUC2022) | 58.2 | 69.1 | 63.2 | 23.3 | 18.1 |
| HarDNet-DFUS (Finetuned) | 67.9 | 79.1 | 73.1 | 36.0 | 30.0 |
| Mask R-CNN | 66.5 | 82.2 | 73.5 | 39.5 | 33.5 |
| MobileNet | 64.6 | 81.7 | 72.2 | 30.4 | 23.6 |
| U-Net | 52.1 | 80.4 | 62.3 | 21.9 | 16.5 |
| SegNet | 57.3 | 63.0 | 60.0 | 18.3 | 13.5 |
| Our Model | **86.8** | **83.4** | **85.1** | **54.4** | **45.4** |

TABLE IV
MODEL PERFORMANCE TRAINED AND TESTED ON DIFFERENT DATASET

| mDice | Training Dataset | | | | | |
|---|---|---|---|---|---|---|
| | FUSeg | AZH | Medetec | Combination | Our Dataset | All Combo |
| Test on FUSeg | 87.0 | 80.4 | 60.3 | 86.8 | 70.7 | 86.5 |
| Test on AZH | 74.4 | 87.5 | 75.7 | 87.5 | 71.5 | 87.0 |
| Test on Medetec | 76.2 | 76.8 | 97.7 | 97.0 | 84.0 | 96.3 |
| Test on Combination | 80.2 | 84.0 | 69.1 | 87.4 | 71.4 | 87.1 |
| Test on Our Dataset | 27.0 | 18.9 | 27.8 | 20.8 | 54.4 | 61.5 |

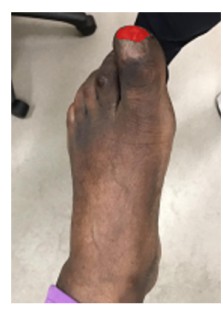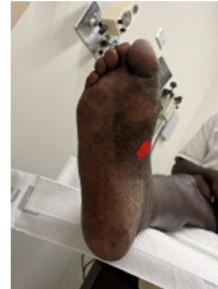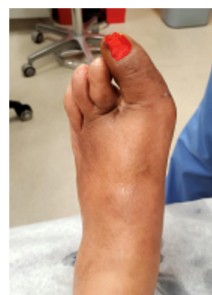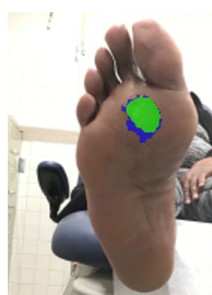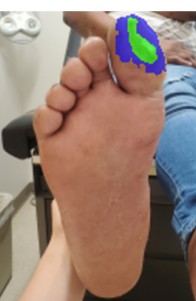

Fig. 3. Examples of model failure cases showing false positive (red), false negative (blue), and correct (green) segmentation masks.

additional datasets. Increasing dataset size will be imperative, as this would help address potential biases, increase model generalizability, and better capture patient diversity. Due to the scarcity of specialized images in dermatologic datasets, future work could also explore self-supervised learning to leverage unlabeled data, enhancing model robustness. Finally, clinical deployments could evaluate clinical impacts of model performance, e.g., patient engagement, clinical decision-making.

Our work assessed the ability of state-of-the-art models trained on other wound datasets (predominantly collected from white patients) to generalize to our dataset collected predominantly from patients of color. It is clear from our experimental results that larger, more diverse datasets will be crucial for the next generation of wound recognition models. In support of more equitable ulcer recognition models, we have made available both the full image dataset (3,362 images) and the computational models utilized in this work: https://github.com/tploetz/dfu-recognition. To our knowledge, this is the first available image dataset of ulcers predominantly collected from patients of color. As automated wound recognition may improve clinician delivery of care, it is vital to advance equity for those who face the greatest disease burden.

ACKNOWLEDGMENT

We would like to thank our 252 participants and the image annotation team at the Georgia Tech Ubicomp Health & Wellness Lab. This work was supported by the American Diabetes Association Grant 11-22-ICTSHD-09, the GT-Emory AI Humanity seed grant, and NIH award DK124647. The first two authors contributed equally to this work.

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
