# OpenReview forum: "Towards More Equitable Ulcer Recognition Models: A Dataset of Naturalistic Foot Images from People of Color Living with Diabetes"
_IEEE.org/EMBS/BHI/2025/Conference — BHI 2025_

### Official Review · Reviewer_LgjY · 2025-07-17
**Towards More Equitable Ulcer Recognition Models: A Dataset of Naturalistic Foot Images from People of Color Living with Diabetes**

**Confidence:** 5
**Clarity Of Writing:** good
**Clinical Significance:** fair
**Methodological Novelty:** poor
**Overall Rating:** 4
**Final Rating:** 7

**Experiments And Results:**

poor

**Questions For The Authors:**

1.	How about the image resolution? It is mentioned neither in the data description nor anywhere.
2.	How many images are there from each participant?
3.	Is there any institutional IRB that was approved before collecting data?This part is essential and missing in the paper.
4.	Patients’ demographic information needs to be mentioned (Ethnicity, race).
5.	There is no clear information regarding training and test dataset.
6.	How many epoch or batch sizes?
7.	It would be great to show the training and validation curves.
8.	At least some basic and common model parameters should be discussed. Fig. 2 need more details regarding Encoder, RFB, and Decoder.

**Strengths:**

The idea was nice.

**Summary Of The Paper:**

This paper worked on creating a repository of diabetic foot ulcer images that have 3362 foot images from 256 patients. This image dataset can be useful for other researchers to explore new insights.
This paper also tried to detect diabetic ulcers using their dataset. However, there are some major and minor concerns regarding Methods and data.

**Weaknesses:**

The modeling and data collection was not described well. Need major revision to improve the paper quality in terms of methods for detecting ulcers from images. More details and questions are given in “Questions for the authors”.

---

### Official Review · Reviewer_QK3u · 2025-07-17
**Towards More Equitable Ulcer Recognition Models: A Dataset of Naturalistic Foot Images from People of Color Living with Diabetes**

**Confidence:** 4
**Clarity Of Writing:** excellent
**Clinical Significance:** great
**Methodological Novelty:** great
**Overall Rating:** 7
**Final Rating:** 7

**Experiments And Results:**

good

**Questions For The Authors:**

1. What is the resolution and preprocessing pipeline for images? Were images resized, normalized, or enhanced in any way before training? These preprocessing steps can significantly influence model performance, particularly for small target regions.

2. Did you try class-balancing techniques beyond data augmentation? The dataset is imbalanced, with only ~34% of images containing lesions. Were any class-weighting methods, sampling strategies, or custom losses (e.g., focal loss) considered?

3. Why was SAM2-UNet chosen over other foundation models for fine-tuning? Given the growing number of pretrained segmentation backbones (e.g., MedSAM, SegFormer), did you compare alternatives?

**Strengths:**

Novel Dataset: The authors present the first publicly available DFU image dataset predominantly collected from people of color in naturalistic settings, capturing the full foot and a wide variety of angles. This dataset fills a critical gap in current medical imaging resources.

Realistic Imaging Conditions: Unlike many existing datasets captured in controlled environments, this dataset was collected in routine clinical conditions using smartphones and iPads. This improves ecological validity and supports future applications in remote monitoring and mobile health tools.

Comprehensive Evaluation: The study evaluates five strong baseline models and thoroughly compares performance across datasets, revealing substantial domain gaps and exposing how existing models fail to generalize to diverse populations.

Methodological Innovation: The proposed ulcer recognition pipeline, which integrates YOLOv12-X and SAM2-UNet, is thoughtfully designed and significantly improves segmentation performance on this challenging dataset.

**Summary Of The Paper:**

The authors introduced the first large-scale dataset of diabetic foot images collected predominantly from patients of color in naturalistic clinical settings. They demonstrated that state-of-the-art DFU segmentation models, which previously achieved high performance on existing datasets, performed significantly worse on their dataset. To address this performance gap, they proposed a novel pipeline combining YOLOv12-X for ulcer detection and SAM2-UNet for segmentation. This approach outperformed all baselines on their dataset.

**Weaknesses:**

Limited Generalizability:
While the dataset focuses on people of color, it is predominantly composed of non-Hispanic Black patients. The authors acknowledge this, but future work would benefit from expanding the demographic diversity (e.g., Latinx, Native American, Southeast Asian) to ensure broader applicability across underserved communities.

Segmentation Performance Still Limited:
Despite outperforming prior models, the proposed pipeline still achieves only 54.4% Dice and 45.4% IoU, which are relatively low compared to performance on standard datasets. This limits immediate clinical utility. Further architectural innovation or data augmentation strategies could help this performance gap.

Lack of Ablation Study:
The proposed pipeline includes multiple components (YOLOv12-X detection, SAM2-UNet segmentation, pixel weighting, dynamic thresholding), but there is no ablation study to isolate the contribution of each module. Understanding which components drive performance gains would strengthen the methodological contribution.

Overreliance on Augmentation to Handle Class Imbalance:
The dataset contains only ~34% images with “areas of concern,” and the authors rely heavily on transformation-based augmentation. However, augmentation alone may not fully address the challenges of small lesion segmentation. Alternative techniques such as focal loss, class rebalancing, or semi-supervised learning could be considered.

Missing Error Analysis or Failure Cases:
The paper does not provide qualitative error analysis or visual examples of segmentation failures. This limits the reader’s ability to understand where and why models fail, especially in cases of poor lighting, atypical lesion appearance, or subtle pre-ulcerative changes

---

### Official Review · Reviewer_o5hb · 2025-07-18
**A timely and impactful study introducing a diabetic foot ulcer image dataset predominantly from patients of color, highlighting existing model inequities and proposing a pipeline for more equitable ulcer recognition.**

**Confidence:** 4
**Clarity Of Writing:** great
**Clinical Significance:** great
**Methodological Novelty:** good
**Overall Rating:** 7

**Experiments And Results:**

great

**Questions For The Authors:**

Could you provide an ablation study showing the contributions of YOLOv12-X and SAM2-UNet independently? This would help isolate which component drives the performance gains.

Have you evaluated inter-rater agreement on the annotations? Understanding annotation variability could impact model trustworthiness.

How representative is the dataset across skin tones within communities of color? This could affect generalizability and future extensions.

Have you considered data or task augmentations (e.g., GANs or self-supervised learning) to boost performance? Might help address the small lesion size challenge.

What steps can be taken to further improve segmentation performance beyond 54.4% Dice? This is still low for clinical-grade models. Suggestions for post-processing or hybrid methods would be valuable.

**Strengths:**

High societal impact: The paper tackles algorithmic bias in wound care and contributes toward more inclusive health AI.

Dataset contribution: A unique, well-documented dataset of images from patients of color, collected under realistic clinical conditions.

Thorough evaluation: Strong experimental framework comparing baseline and fine-tuned models, cross-dataset evaluation, and careful metric selection.

Strong empirical evidence: Demonstrates real-world degradation in model performance on underrepresented populations.

Pipeline performance: Proposed detection + segmentation pipeline significantly outperforms prior models on this challenging data.

Reproducibility: Plans to release both dataset and models, providing a strong foundation for future work.

**Summary Of The Paper:**

This paper introduces a new dataset of 3,362 diabetic foot images collected from 256 patients, primarily patients of color, over a two-year period at a U.S. safety net hospital. It details the data collection and annotation process, emphasizing real-world, naturalistic conditions (e.g., varying lighting, device types, and foot angles). The paper benchmarks existing ulcer recognition models (e.g., Mask R-CNN, HarDNet-DFUS, U-Net) on this dataset and finds a drastic drop in performance compared to results on traditional datasets. To improve performance, the authors propose a new pipeline combining YOLOv12-X for ulcer detection and SAM2-UNet for segmentation. Their model improves classification F1 score to 85.1% and segmentation Dice score to 54.4%, outperforming all baselines. The paper advocates for equitable model development, releasing both the dataset and code upon acceptance.

**Weaknesses:**

Moderate methodological novelty: The core contribution is dataset-centric; the proposed pipeline leverages known components (YOLOv12-X, SAM2-UNet) without significant architectural innovation.

Limited generalizability discussion: While focused on Black patients, there is limited discussion of how the model might generalize to other communities of color (e.g., Latinx, Native American).

Absence of clinical outcome linkage: The paper does not yet evaluate whether the improved model performance translates into better clinical decision-making or patient outcomes.

Limited segmentation robustness: Even the best-performing model (Dice: 54.4%) may fall short of clinical applicability; this warrants further investigation or post-processing refinements.

Missing ablation study: No breakdown is provided of how much each module in the proposed pipeline contributes to performance.

No interpretability discussion: How the model makes decisions, particularly on darker skin tones, could be explored to better align with fairness goals.

---

### Official Review · Reviewer_Udhx · 2025-07-18
**Exposing the Equity Gap: When State-of-the-Art Diabetic Foot Ulcer Detection Fails Communities of Color**

**Confidence:** 3
**Clarity Of Writing:** good
**Clinical Significance:** great
**Methodological Novelty:** fair
**Overall Rating:** 7

**Experiments And Results:**

good

**Questions For The Authors:**

Questions:
1. The 54.4% Dice score is still quite low for clinical deployment. What performance threshold do you believe would be clinically acceptable, and what's your roadmap to get there?
2. How confident are you that the performance gap is primarily due to skin tone rather than other confounding factors like wound characteristics, imaging conditions, or hospital-specific practices?
3. How do you plan to expand beyond the predominantly Black patient population to include other communities of color (Hispanic, Native American, Asian, etc.)?

**Strengths:**

What I appreciate about this work is how methodical the data collection was. Two years of collaboration with a safety-net hospital, getting informed consent, compensating participants fairly, and capturing images with smartphones in real clinical settings rather than controlled lab conditions - this feels like research done the right way. The fact that they included the whole foot rather than just close-ups of wounds is also smart from a practical deployment perspective.

Findings like these can be very useful for companies like Empo Health which are using CV to detect foot ulcers - authors should consider sharing these findings with them.

The technical contribution is solid if not groundbreaking. The YOLOv12-X + SAM2-UNet pipeline makes sense, and the performance improvements are meaningful (54.4% vs 39.5% Dice score). More importantly, they show that models trained on their diverse dataset actually generalize better to existing datasets than the reverse, which is a compelling argument for inclusive training data.

What really stands out is how this exposes the blind spots in our field. We've been celebrating 90%+ accuracy on datasets that apparently don't represent the populations most affected by diabetic foot ulcers. The authors have done the community a service by releasing this dataset publicly - it should become a standard benchmark for anyone working on wound detection.

**Summary Of The Paper:**

Looking at this paper, I have to say the authors have tackled a genuinely important problem that's been largely overlooked in the computer vision community. The core finding is pretty striking - state-of-the-art diabetic foot ulcer detection models that supposedly achieve 90%+ accuracy completely fall apart when tested on images from Black patients, dropping to under 40% performance. That's not just a technical limitation; it's a serious equity issue.
This feels like important foundational work that will hopefully catalyze more inclusive dataset development across medical AI. The technical contributions are incremental but appropriate, and the broader impact could be substantial.

**Weaknesses:**

My main concerns are around generalizability - this is primarily Black patients from one hospital system in the South. While the authors acknowledge this limitation, it does raise questions about how well this extends to other communities of color. The performance is also still quite low compared to what we'd want for clinical deployment, though that may be more realistic given the challenging naturalistic imaging conditions.